# Unifying Knowledge from Diverse Datasets to Enhance Spatial-Temporal Modeling: A Granularity-Adaptive Geographical Embedding Approach

Zhigaoyuan Wang [1]   Ying Sun* [1]   Hengshu Zhu [2 3]

## Abstract

Spatio-temporal forecasting provides potential for discovering evolutionary patterns in geographical scientific data. However, geographical scientific datasets are often manually collected across studies, resulting in limited time spans and data scales. This hinders existing methods that rely on rich historical data for individual entities. In this paper, we argue that heterogeneous datasets from different studies can provide complementary insights into the same underlying system, helping improve predictions for geographical entities with limited historical data. To this end, we propose a Segment Quadtree Geographical Embedding Framework (SQGEF). SQGEF integrates knowledge from datasets with varied target entities, time spans, and observation variables to learn unified representations for multi-granularity entities—including those absent during training. Specifically, we propose a novel data structure, Segment Quadtree, that flexibly accommodates entities of varying granularities. SQGEF not only captures multi-level interactions from grid data but also extracts nested relationships and human-defined boundaries from diverse entities, enabling a comprehensive understanding of complex geographical structures. Experiments on real-world datasets demonstrate that SQGEF effectively represents unseen geographical entities and enhances performance for various models.

## 1. Introduction

Spatial-temporal data and its analysis are crucial across various fields. For instance, analyzing trends in spatio-temporal carbon emission data for geographic entities (Peters et al., 2012) is essential for developing effective environmental strategies. Additionally, examining changes in their global spatial distribution aids in improving climate modeling.

In the past decade, extensive efforts have been made in spatio-temporal forecasting. In time series forecasting, current methods treat the observations of each target entity as an time series, leveraging historical data to uncover temporal patterns, such as frequencies (Zhou et al., 2022) and trends (Wu et al., 2022), within individual entities for downstream forecasting. In contrast, spatio-temporal forecasting goes beyond using each entity's own history by further capturing interactions between multiple target entities. For example, ST-ResNet (Zhang et al., 2017) employs CNNs to model relationships between geographically nearby entities, while GNN-based methods (Wu et al., 2019b) extend this to capture both local and long-distance dependencies across entities. These approaches, grounded in target entities' histories, have proven effective in forecasting tasks such as electricity demand and traffic flow prediction.

However, traditional spatio-temporal methods heavily rely on abundant historical data, which is often limited in scientific studies due to the high cost of data collection. Many scientific datasets are gathered through expensive specialized equipment, which leads to data being collected from only a small number of targets. Additionally, time-consuming survey methods restrict the time span of the datasets. As a result, spatio-temporal methods struggle to capture entity relationships from such sparse datasets.

In scientific research scenarios, although individual studies have limited data, numerous related studies exist, and their data are inherently connected. Different datasets collect different entities across various granularities, yet they share inherent geographical relationships. For example, provincial-level carbon emission data provides a macro-level reflection of the emissions patterns observed at the city level, offering valuable prior information when city-level data is scarce. Likewise, neighboring cities often experience similar en-

---

Corresponding author: * [1]AI Thrust, HKUST(GZ), Guangzhou, China [2]Computer Network Information Center, Chinese Academy of Sciences, Beijing, China [3]University of Chinese Academy of Sciences, Beijing, China. Correspondence to: Ying Sun <yings@hkust-gz.edu.cn>.

*Proceedings of the $42^{nd}$ International Conference on Machine Learning*, Vancouver, Canada. PMLR 267, 2025. Copyright 2025 by the author(s).

vironmental conditions, making one city's data useful for inferring patterns in the other. Second, different metrics within a unified industrial system are often correlated. For instance, carbon emissions and energy consumption, though sourced from different studies, all reflect energy usage dynamics and can compensate for missing observations. Consequently, the information within these different datasets is inherently complementary. Leveraging these cross-dataset relationships enables the integration of entity interactions across datasets while capturing their hierarchical structure and geographical dependencies. Therefore, we argue that: **heterogeneous datasets from different studies can provide complementary insights into the same underlying system, thereby improving predictions for geographical entities with limited historical data.**

However, integrating heterogeneous datasets to model unseen entities presents several challenges. First, the target task dataset may contain entities that have not appear in the avaliable datasets, obtaining relevant information for unseen entities is challenging. Second, granularity of different datasets can largely differ, each granularity offering unique interactions. For example, country-level interactions in national datasets differ from city-level interactions in city datasets. Finding a way to utilize these heterogeneous datasets to model unified complex interactions is a significant challenge. Third, datasets can have different data structures, such as entity-based and grid-based. Entity-based datasets present interactions within human-defined boundaries with administrative knowledge, while grid-based datasets present interactions within naturally divided units of a region with geographical information. Extracting and fusing relationships from different data structures to create an integrated model is another key challenge.

To address these challenges, we propose Segment Quadtree Geographical Embedding Framework (SQGEF) that utilizes heterogeneous datasets to achieve a unified representation of different granularity entities within a region, even if they are not present in the training dataset. This framework is built on a novel data structure, the Segment Quadtree, and two innovative learning methods tailored to train this structure on entity and grid datasets, respectively. (1) The Segment Quadtree provides a hierarchical representation of the region, in contrast to single-layer grid-based partitioning that treats all regions uniformly. It offers the flexibility to represent entities and grids of any granularity while explicitly capturing nested relationships between nodes, enabling multi-level information storage and aggregation. (2) Hierarchical Grid-Based Learning captures interactions at multiple levels from grid datasets, incorporating global interactions for child nodes within the Segment Quadtree. (3) Geographical Entity-Based Learning aggregates grids within entities to extract nested relationships and human-defined boundaries information across different granularity entities.

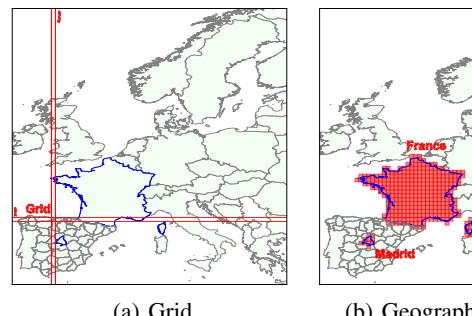

(a) Grid       (b) Geographical Entity

Figure 1: (a) Grid $\mathcal{G}_{i,j}$ in the European region. (b) Geographical entity consisting of grids France at the country granularity and Madrid at the province granularity.

We conduct experiments on country, province, and city-granularity forecasting tasks across different regions. The results demonstrate that our method effectively models geographical entities of varying granularities from different regions, benefiting both timeseries and spatial-temporal models. Further experiments prove the robustness of our model, and the embeddings we learn effectively reflect the relationships of entities in the real world.

The key contributions of this work are as follows:

- To our knowledge, we are among the first to propose a novel framework that can integrate different heterogeneous datasets to address the data scarcity issue in scientific datasets.
- We propose a novel embedding method that can represent different granularity entities within a region, even if they are not present in the training dataset.
- We conduct comprehensive experiments demonstrating that our method effectively represents unseen geographical entities across all granularities and regions, benefiting a range of models and showcasing its versatility and robustness.

## 2. Preliminaries

### 2.1. Problem Formulation

**Definition 1: Grid**: In this study, we partition the region containing all geographical entities into an $\mathcal{I} \times \mathcal{J}$ grid map based on longitude and latitude, where each grid cell $\mathcal{G}_{i,j}$ represents a specific area within the current region, as illustrated in Figure 1(a).

**Definition 2: Geographical Entity**: A geographical entity $\mathcal{E}_k$ is represented as a set of grid cells. Specifically, if a grid cell $\mathcal{G}_{i,j}$ is part of the geographical entity $\mathcal{E}_k$, then $\mathcal{G}_{i,j} \in \mathcal{E}_k$. Therefore, a geographical entity $\mathcal{E}_k$ can be defined as: $\mathcal{E}_k = \{\mathcal{G}_{i,j} \mid (i,j) \in \mathcal{I}_k \times \mathcal{J}_k\}$ where $\mathcal{I}_k \subseteq \{1, \ldots, \mathcal{I}\}$ and $\mathcal{J}_k \subseteq \{1, \ldots, \mathcal{J}\}$ denote the indices of the grid cells that belong to the entity $\mathcal{E}_k$, as shown in Figure 1(b).

In this paper, we aim to forecast the multivariate time series for a given set of geographical entities, denoted as

$\mathcal{E}^{test} = \{\mathcal{E}_1^{test}, \mathcal{E}_2^{test}, \ldots, \mathcal{E}_{N^{test}}^{test}\}$. Let $\mathbf{x}_t \in \mathbb{R}^N$ represent the values of $N^{test}$ geographical entities at time step $t$, where $x_t[i] \in \mathbb{R}$ denotes the value of the $i$-th geographical entity at time step $t$. Given a sequence of historical observations over $L_X$ time steps, $\mathbf{X} = \{\mathbf{x}_{t_1}, \mathbf{x}_{t_2}, \ldots, \mathbf{x}_{t_{L_X}}\}$, our objective is to predict the future values $\mathbf{Y} = \{\mathbf{x}_{t_{L_X+1}}, \mathbf{x}_{t_{L_X+2}}, \ldots, \mathbf{x}_{t_{L_X+L_Y}}\}$ of these geographical entities based on the previous $L_X$ time steps and additional relevant information.

The additional information includes: 1. **Grid Dataset**: Let $\mathbf{G}_t \in \mathbb{R}^{\mathcal{I} \times \mathcal{J}}$ represent the values of the grid cells at time step $t$, where $\mathbf{G}_t[i,j]$ denotes the value of the grid cell $\mathcal{G}_{i,j}$ at time step $t$. Given a sequence of historical observations over $L_X$ time steps, $\mathbf{G} = \{\mathbf{G}_{t_1}, \mathbf{G}_{t_2}, \ldots, \mathbf{G}_{t_{L_X}}\}$. 2. **Geographical Entity Dataset**: Let $\mathcal{E}^{train}$ represent the set of geographical entities in the training dataset, where $\mathcal{E}^{train} = \mathcal{E}_1^{train}, \mathcal{E}_2^{train}, \ldots, \mathcal{E}_{N^{train}}^{train}$, and $N^{train}$ denotes the number of geographical entities in the training dataset. Each entity $\mathcal{E}_k^{train}$ has an associated time series $\mathbf{X}_k^{train} = x_k^{train}[t_1], x_k^{train}[t_2], \ldots, x_k^{train}[t_{L_X}]$ for historical observations.

It is important to note that different Geographical Entity datasets have entirely *different* sets of geographical entities, i.e., $\mathcal{E}^{train} \cap \mathcal{E}^{test} = \emptyset$. The entities in the target dataset $\mathcal{E}^{test}$ *will not appear in training datasets* $\mathcal{E}^{train}$, emphasizing the challenge of transferring knowledge across datasets to unseen entities. By integrating these heterogeneous datasets, our model effectively predict the multivariate time series for the given set of geographical entities.

### 2.2. Framework Overview

Our framework contains a data structure and two learning methods to train it, as shown in Figure 2. Before presenting the details, we first introduce the pipeline of how to train the model and then elaborate on the motivations and advancements behind their design.

**Segment Quadtree Embedding.** To represent entities at different granularities, we use a hierarchical representation of a region by dividing it into multiple levels, with each child node representing a quarter division of its parent. This approach ensures that nodes represent all granularities within the region. Each node in the Segment Quadtree corresponds to a specific granularity level, capturing the interactions and information relevant to that level.

An entity's variation is influenced by higher-level variations, so the Segment Quadtree is trained from the top down. This process begins with a high-level representation and progressively models more fine-grained representations at lower levels. During this training, lower-level nodes are trained simultaneously with higher-level information, allowing each node to incorporate both the global context from its parent node and the local information specific to itself.

**Hierarchical Grid-Based Learning.** To capture interactions across multiple levels in grid datasets, instead of training the model solely on the original granularity, which focuses only on interactions at a specific level, our approach re-divides datasets into different granularities for training. By training child nodes together with their ancestor nodes on these re-divided datasets, our method incorporates global interactions, enabling each node to understand and capture interactions at various levels.

**Geographical Entity-Based Learning.** Various entities of different granularities from subregions within a region exhibit distinct boundaries and nested relationships that define meaningful groupings and interactions within the grids. Ignoring these aspects would result in a loss of administrative knowledge regarding interactions within and between grids of human-defined entities. Focusing solely on the relationships between grids and their nearby grids overlooks the inherent group information among grids within the same entity. Our approach addresses this limitation by training grids within the same entity together. This helps the model learn both the nested relationships and the human-defined boundaries within different entities. Each entity can be represented by nodes at different levels within the Segment Quadtree, facilitating communication across levels and parent nodes.

## 3. Method

In this section, we present the details of SQGEF.

### 3.1. Segment Quadtree Embedding

To effectively model geographical regions, it is essential to represent each grid within a defined area, as these grids serve as the fundamental units for understanding spatial interactions. By modeling each grid, we can capture localized phenomena and their time series data, providing insights into how different geographical areas interact over time. However, treating each grid in isolation presents significant drawbacks. This approach neglects the hierarchical relationships that exist between different granularity levels of geographical entities, such as cities, provinces, and countries, leading to a fragmented understanding of spatial dynamics.

To address these limitations, we employ a segment quadtree structure, which allows us to represent the region in a hierarchical manner. Formally, let $\mathcal{T}$ denote the segment quadtree. The root node $\mathcal{T}_{0,0}$ covers the entire $\mathcal{I} \times \mathcal{J}$ grid region, and each node $\mathcal{T}_{i,j}$ at level $i$ can be divided into four child nodes $\mathcal{T}_{i+1,4j+1}$, $\mathcal{T}_{i+1,4j+2}$, $\mathcal{T}_{i+1,4j+3}$, and $\mathcal{T}_{i+1,4j+4}$ at level $i+1$, corresponding to the northwest, northeast, southwest, and southeast quadrants, respectively. Each node $\mathcal{T}_{i+1,4j+k}$ (where $k \in \{1, 2, 3, 4\}$) contains the grid cells $\mathcal{G}_{a,b}$ where $a$ and $b$ correspond to the appropriate size of the region: $[2^i \cdot \frac{\mathcal{I}}{2^i}, 2^{i+1} \cdot \frac{\mathcal{I}}{2^i} - 1]$ or $[2^{i+1} \cdot \frac{\mathcal{I}}{2^i}, 2^{i+2} \cdot \frac{\mathcal{I}}{2^i} - 1]$ for $a$, and $[2^j \cdot \frac{\mathcal{J}}{2^j}, 2^{j+1} \cdot \frac{\mathcal{J}}{2^j} - 1]$ or $[2^{j+1} \cdot \frac{\mathcal{J}}{2^j}, 2^{j+2} \cdot \frac{\mathcal{J}}{2^j} - 1]$ for $b$, depending on the quadrant. This hierarchical structure ensures that different granularity representations are learned, preserving the hierarchical nature of geographical entities. Extracting features from the segment quadtree and

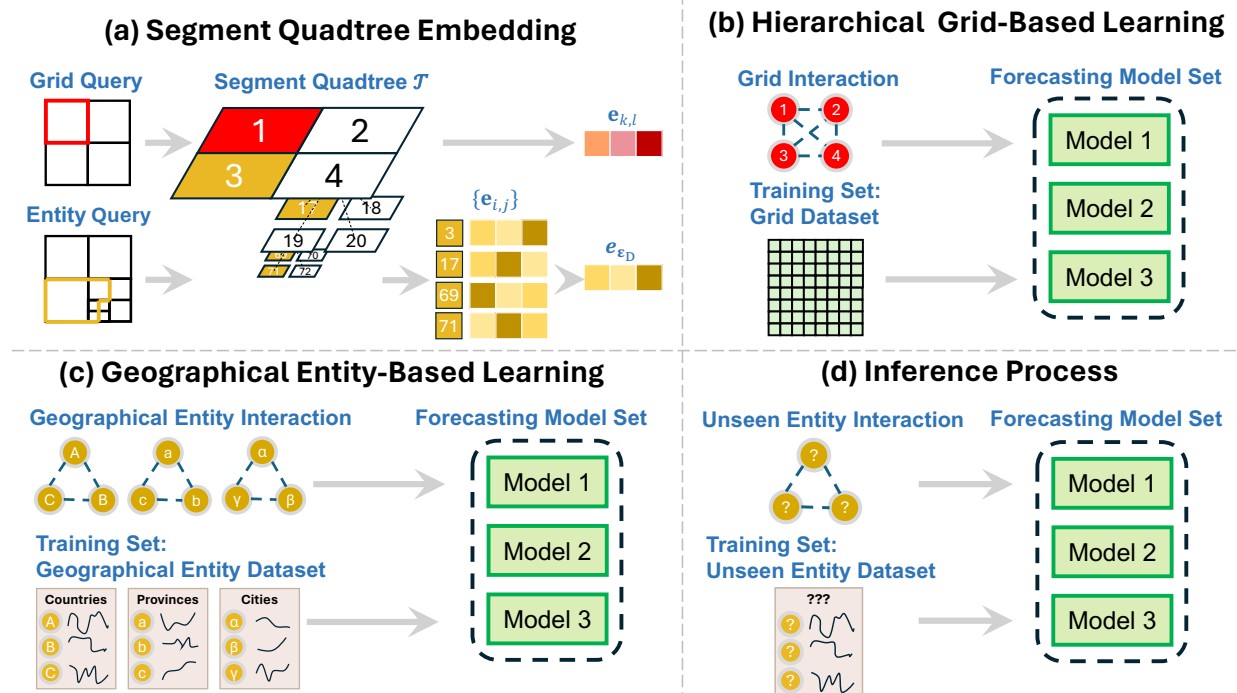

Figure 2: Overview of SQGEF

obtaining the embeddings allows us to construct the relations of any subregions or entities, enhancing time series forecasting methods. This approach captures interactions at multiple levels and provides a comprehensive understanding of spatial dynamics, benefiting forecasting on target datasets.

### 3.2. Hierarchical Grid-Based Learning

In time series methods, the forecast of multivariate time series for a given set of geographical entities, denoted as $\mathcal{E} = \mathcal{E}_1^{test}, \mathcal{E}_2^{test}, \ldots, \mathcal{E}_N^{test}$, can be enhanced by leveraging a graph structure $\mathcal{G} \in \mathbb{R}^{N \times N}$, which captures the contextual relationships between variables. This graph-based approach provides additional context, improving the model's ability to capture interactions between variables. Here, we introduce the construction of $\mathcal{G}$ between each grid.

To effectively train each node and its children together, we construct the graph for all nodes at the same level using the embeddings of each node and its children to calculate the adjacency matrix. For each node $\mathcal{T}_{i,j}$ at level $i$, we compute the mean embedding $\mathbf{m}_{i,j}$, which represents the node $\mathcal{T}_{i,j}$ and its children. Let $\mathcal{C}(\mathcal{T}_{i,j})$ be the set of children of node $\mathcal{T}_{i,j}$. The mean embedding $\mathbf{m}_{i,j}$ is then given by:

$$\mathbf{m}_{i,j} = \frac{1}{|\{\mathcal{T}_{i,j}\} \cup \mathcal{C}(\mathcal{T}_{i,j})|} \sum_{\mathcal{T}_{k,l} \in \{\mathcal{T}_{i,j}\} \cup \mathcal{C}(\mathcal{T}_{i,j})} \mathbf{e}_{k,l}, \quad (1)$$

where $\{\mathcal{T}_{i,j}\} \cup \mathcal{C}(\mathcal{T}_{i,j})$ contains the node $\mathcal{T}_{i,j}$ and its children, and $|\{\mathcal{T}_{i,j}\} \cup \mathcal{C}(\mathcal{T}_{i,j})|$ is the cardinality of this set. Using the mean embeddings $\mathbf{m}_{i,j}$, the adjacency matrix $\mathcal{A}$ for the graph $\mathcal{G}$ is constructed. The adjacency matrix

is calculated based on the cosine similarity between the mean embeddings of different nodes. For instance, the entry $\mathcal{A}_{(i,j),(i',j')}$ represents the relationship between node $\mathcal{T}_{i,j}$ and node $\mathcal{T}_{i',j'}$ and is computed as:

$$\mathcal{A}_{(i,j),(i',j')} = \frac{\mathbf{m}_{i,j} \cdot \mathbf{m}_{i',j'}}{\|\mathbf{m}_{i,j}\|\|\mathbf{m}_{i',j'}\|}, \quad (2)$$

where $\mathbf{m}_{i,j} \cdot \mathbf{m}_{i',j'}$ is the dot product of the mean embeddings and $\|\mathbf{m}_{i,j}\|$ and $\|\mathbf{m}_{i',j'}\|$ are their magnitudes.

Several models $m_1, m_2, m_3, \ldots$ are considered, each taking as input the data $\mathbf{X}$ and the graph $\mathcal{G}$ with shared weights. Let $\mathcal{L}_i$ denote the loss for model $m_i$.

To optimize the performance across all models, we train each model in turn. For each model $m_i$, the loss $\mathcal{L}_i$ is defined using the $\ell_1$ loss between the output and the target:

$$\mathcal{L}_i(\mathbf{X}, \mathcal{G}; \Theta_i, \mathbf{W}) = \sum_{t=1}^{T} |\hat{y}_{i,t} - y_{i,t}|, \quad (3)$$

where $\hat{y}_{i,t}$ is predicted value and $y_{i,t}$ is the target value for model $m_i$ at time step $t$ and $\mathbf{W}$ represents the shared weights to construct the graph $\mathcal{G}$ with adjacency matrix $\mathcal{A}$.

The training process involves minimizing the loss for each model sequentially. Each model $m_i$ has its specific parameters $\Theta_i$, but they share the weights of the same graph $\mathcal{G}$. The combined optimization objective, taking into account the shared weights of the graph, is:

$$\mathcal{G}^* = \arg\min_{\mathcal{G}_i} \mathcal{L}_i(\mathbf{X}, \mathcal{G}; \Theta_i) \quad for \ i = 1, 2, \ldots, n, \quad (4)$$

where $\{\Theta_i\}$ represents the parameters of all models.

By iteratively training each model on multiple grid datasets, the shared graph $\mathcal{G}$ is refined with its level nodes and their child nodes from Segment Quadtree.

### 3.3. Geographical Entity-Based Learning

Geographical entities are defined by distinct administrative boundaries that dictate meaningful groupings and nested relations within the data. To effectively capture these boundaries and interactions of grids within their boundaries, we train the Quadtree $\mathcal{T}$ on Geographical Entity Datasets.

For each geographical entity $\mathcal{E}_D$ in the dataset $\mathcal{E}^{train}$, the grid layout is represented by a binary matrix $\mathcal{M}_{\mathcal{E}_D} \in \{0,1\}^{\mathcal{I} \times \mathcal{J}}$, where 1 indicates grid presence and 0 indicates absence. To extract the entity representation from the grid representation using the quadtree traversal, we traverse from the root node $\mathcal{T}_{0,0}$ downward. During traversal, if a node $\mathcal{T}_{i,j}$ overlaps with any 1 in $\mathcal{M}_{\mathcal{E}_D}$, it is added to the set $\mathcal{Q}_{\mathcal{E}_D}$, and traversal of its child nodes is halted. This approach ensures that nodes $\mathcal{T}_{i,j} \in \mathcal{Q}_{\mathcal{E}_D}$ at various levels $i$ capture different aspects of $\mathcal{E}_D$'s spatial structure, effectively representing the spatial layout of $\mathcal{E}_D$ at different granularities.

The embedding $\mathbf{e}_{\mathcal{E}_D}$ for $\mathcal{E}_D$ is obtained by aggregating the embeddings of nodes $\mathcal{T}_{i,j} \in \mathcal{Q}_{\mathcal{E}_D}$, weighted by their level $i$. Specifically, the embedding is computed as:

$$\mathbf{e}_{\mathcal{E}_D} = \sum_{\mathcal{T}_{i,j} \in \mathcal{Q}_{\mathcal{E}_D}} \frac{\mathbf{e}_{i,j}}{2^i}, \tag{5}$$

where $\mathbf{e}_{i,j}$ is the embedding of node $\mathcal{T}_{i,j}$. This hierarchical aggregation ensures that $\mathbf{e}_{\mathcal{E}_D}$ captures information from different levels of $\mathcal{T}$, allowing nodes from various levels to effectively communicate their information. By leveraging the embeddings $\mathbf{e}_{\mathcal{E}_D}$ obtained for each geographical entity $\mathcal{E}_D$, we construct a graph $\mathcal{G}_{\mathcal{E}_D}$ that captures the contextual relationships between these entities. Each node in $\mathcal{G}_{\mathcal{E}_D}$ represents a geographical entity, and edges between nodes encode relationships based on their embeddings.

Similar to Grid-Based Learning, we train sharing weights of the same graph $\mathcal{G}$ on several models $m_1, m_2, m_3, \ldots$ in turn. Unlike Grid-Based Learning, which focuses on single-level grid cells, geographical entity-based learning aggregates information across hierarchical nodes of the Segment Quadtree and captures longer-distance interactions between nodes from different parent nodes. This approach enables the model to capture dependencies and interactions that span various geographical scales and political boundaries.

### 3.4. Model Training and Inference

Having defined two distinct learning approaches, we now proceed with the overall training process for the segment quadtree and inference on downstream datasets.

3.4.1. TRAINING PROCESS

The Grid Dataset serves as the initial training data for constructing the segment quadtree $\mathcal{T}$. This dataset provides spatial information organized into a grid structure, essential

for initializing and optimizing the hierarchical segmentation within $\mathcal{T}$. After training with the Grid Dataset, we proceed to train using the Geographical Entity Dataset. Each dataset contains entities at varying granularities, such as country, province, and city datasets, allowing the segment quadtree to extract hierarchical grouping information at different levels. This dataset enables the refinement of the segment quadtree $\mathcal{T}$ to capture specific hierarchical representations tailored to geographical entities.

3.4.2. INFERENCE PROCESS

Once the segment quadtree $\mathcal{T}$ has been trained on both types of datasets, we use it to enhance any models for inference on target datasets. Given the boundaries of the set of geographical entities appearing in the downstream dataset $\mathcal{E}^{test} = \{\mathcal{E}_1^{test}, \mathcal{E}_2^{test}, \ldots, \mathcal{E}_N^{test}\}$, we query $\mathcal{T}$ to get representations for all entities and build the interaction graph $\mathcal{G}$. We train the model on historical observations $\mathbf{X}$ over $L_X$ time steps. The interaction graph $\mathcal{G}$ and the inference model are jointly trained on this historical data to fit the downstream dataset with pretrained knowledge. After training, we perform inference on the test set $\mathbf{Y}$, where $\mathbf{Y}$ consists of time steps following the historical data. The trained model leverages the pretrained representations from $\mathcal{T}$ and task-specific relations from the downstream dataset to predict results for these future time steps.

## 4. Theoretical Analysis

In this section, we briefly analyze how the segment quadtree representation regulates model complexity to better capture entity relationships, the complete proof is provided in Appendix A.5. We prove that the SQGEF yields a tighter generalization error bound compared to the traditional Grid-based Partitioning Embedding using Rademacher complexity. We compare two embedding methods for an entity with $k$ grids region: **Grid-based Partitioning Embedding**: Fuses $k$ grid embeddings using $k$ embeddings. The hypothesis class is denoted $\mathcal{H}_{\text{grid}}$. **Segment Quadtree Embedding**: Constructs a segment tree over the grids of region, fusing $m = O(\log k)$ node embeddings to represent the entity. The hypothesis class is denoted $\mathcal{H}_{\text{seg}}$.

The Grid-based Partitioning Embedding method employs an MLP to map $k$ embeddings, each in $\mathbb{R}^d$, into a single $d$-dimensional embedding. The input is a concatenation of $k$ vectors (dimension $kd$), with $L$ hidden layers and parameter count $p$ scaling with $kd$. The Rademacher complexity is bounded as (Bartlett et al., 2017):

$$\hat{\mathcal{R}}_S(\mathcal{H}_{\text{grid}}) \leq C_1 \cdot \frac{\|W\|^{L+1} \sqrt{p} \sqrt{kd}}{\sqrt{n}}, \tag{6}$$

where $C_1 > 0$ is a constant, $\|W\|$ is the maximum spectral norm of weight matrices, and $p \propto kd$ due to input layer weights.

Similarly, the Segment Quadtree Embedding fuses $m = O(\log k)$ embeddings, each in $\mathbb{R}^d$, into a single $d$-dimensional embedding using an MLP. The input dimension is $md$, with $L$ hidden layers and parameter count $p'$ scaling with $md$. The Rademacher complexity is bounded as:

$$\hat{\mathcal{R}}_S(\mathcal{H}_{\text{seg}}) \leq C_2 \cdot \frac{\|W\|^{L+1}\sqrt{p'}\sqrt{md}}{\sqrt{n}}, \qquad (7)$$

where $C_2 > 0$ is a constant, and $p' \propto md$.

Assume $L$ and $|W| > 1$ are the same for both methods, and the constants $C_1 \approx C_2$ are approximately equal, with $p \propto kd$ and $p' \propto md$. The Rademacher complexities are $\hat{\mathcal{R}}_S(\mathcal{H}_{\text{grid}}) \propto \|W\|^{L+1}\sqrt{kd}\sqrt{p}$ for Grid-based Partitioning Embedding and $\hat{\mathcal{R}}_S(\mathcal{H}_{\text{seg}}) \propto \|W\|^{L+1}\sqrt{md}\sqrt{p'}$ for Segment Quadtree Embedding. Since $m = O(\log k) < k$, and $p' < p$, we have $\sqrt{md}\sqrt{p'} < \sqrt{kd}\sqrt{p}$, thus $\hat{\mathcal{R}}_S(\mathcal{H}_{\text{seg}}) < \hat{\mathcal{R}}_S(\mathcal{H}_{\text{grid}})$, the generalization bound for the Segment Quadtree Embedding is tighter.

## 5. Experiments

Table 1: Experimental Datasets for Training and Testing

| Dataset Name | Data Type | Function |
|---|---|---|
| ODIAC 1km China | Grid | Train |
| ODIAC 1deg China | Grid | Train |
| MEICModel Region | Entity | Train |
| **Carbonmonitor China** | **Entity** | **Test** |
| ODIAC 1km EU | Grid | Train |
| ODIAC 1deg EU | Grid | Train |
| **Carbonmonitor EU** | **Entity** | **Test** |
| ODIAC 1km China | Grid | Train |
| ODIAC 1deg China | Grid | Train |
| Carbonmonitor China | Entity | Train |
| **MEICModel City** | **Entity** | **Test** |

To validate that SQGEF can represent unseen different granularity geographical entities from various regions, we apply it to CO2 emission forecasting tasks for entities at three different granularities: country, province, and city, across two distinct regions—China and Europe. For each region, we utilize multiple grid or geographical entity datasets to train the model and subsequently test it on a geographical entity dataset not included in the training phase. We then analyze the experimental results, demonstrating improvements in accuracy by comparing with various baselines. Additionally, we design different experiments to address the following questions: **Q1:** Can our method effectively model unseen geographical entities of different granularities? How do factors such as the granularity, region of entities, and extreme data scarcity impact performance? **Q2:** Is Segment Quadtree necessary for effective modeling, as opposed to modeling each grid individually? **Q3:** Does Geographical

Entity-Based Learning contribute to improved model performance? **Q4:** How do various hyperparameter settings influence the performance of the model? **Q5:** Does the model accurately learn relationships that reflect the actual interactions between entities in real world? **Q6:** Can our method generalize effectively across datasets from different scientific domains?

### 5.1. Data Description

We set three downstream forecasting tasks with different granularities from two regions for carbon emissions: China Province, Europe Country, and China City. The China City dataset suffers from extreme data scarcity, making it unsuitable for some baseline models. The datasets used for each task are summarized in Table 1, with detailed descriptions provided in the appendix.

### 5.2. Experimental Setup

#### 5.2.1. METRICS

We used mean absolute error (MAE) and mean squared error (MSE) to assess the performance of our model and baselines. For both of these metrics, smaller values indicate better performance.

#### 5.2.2. BASELINES

We selected seven representative baseline methods, falling into two categories: time series models and spatio-temporal models. For the first category, we compare the three most widely utilized time series models: **Informer** (Zhou et al., 2021), **FEDformer** (Zhou et al., 2022), **Autoformer** (Wu et al., 2021), as well as the recent state-of-the-art method **TimesNet** (Wu et al., 2022). For the second category, we compare the three most widely utilized spatial-temporal models: **AGCRN** (Bai et al., 2020), **MTGNN** (Wu et al., 2020), and **GWNet** (Wu et al., 2019a). Detailed descriptions are provided in the appendix.

### 5.3. Overall Performance(Q1)

As shown in Table 2,**w/ Seg** denotes the enhancement of baseline models by SQGEF. SQGEF achieves the best performance across all datasets and generally improves the performance of all baseline methods. This indicates that SQGEF effectively represents entities of different granularities, from countries to cities in various regions, and benefits various kinds of models.

On all datasets, SQGEF shows greater improvements for time series methods compared to spatio-temporal methods. This is because SQGEF helps the time series methods to learn spatial relations, which they were unable to capture before. Additionally, SQGEF can also enhance the performance of spatio-temporal baselines. This indicates that SQGEF learns better representations from the training dataset compared to a completely data-driven approach.

However, the China City dataset is unique because all the cities it contains are obscure and less representative in the

Table 2: Overall performance comparison. The best results among all the models are highlighted in bold.

| China Province | | | Europe Country | | | China City | | |
|---|---|---|---|---|---|---|---|---|
| Method | MSE | MAE | Method | MSE | MAE | Method | MSE | MAE |
| Informer | 0.8890 ± 0.0023 | 0.7329 ± 0.0006 | Informer | 2.0979 ± 0.0027 | 1.0842 ± 0.0003 | Informer | 7.6327 ± 0.1954 | 1.7926 ± 0.0185 |
| w/Seg | 0.7638 ± 0.0011 | 0.6815 ± 0.0001 | w/Seg | 2.0373 ± 0.0004 | 1.0658 ± 0.0001 | w/Seg | 7.3984 ± 0.0239 | 1.7488 ± 0.0069 |
| FEDformer | 0.3588 ± 0.0000 | 0.4753 ± 0.0000 | FEDformer | 2.7053 ± 0.0000 | 1.2411 ± 0.0000 | FEDformer | 6.5412 ± 0.4708 | 1.3191 ± 0.0005 |
| w/Seg | 0.2941 ± 0.0000 | 0.3971 ± 0.0000 | w/Seg | 2.3831 ± 0.0002 | 1.1772 ± 0.0000 | **w/Seg** | **6.2043 ± 0.3618** | **1.1631 ± 0.0006** |
| Autoformer | 0.3242 ± 0.0003 | 0.4490 ± 0.0001 | Autoformer | 2.7713 ± 0.0002 | 1.2554 ± 0.0000 | Autoformer | 7.3596 ± 0.6210 | 1.3848 ± 0.0103 |
| w/Seg | 0.2934 ± 0.0000 | 0.4008 ± 0.0000 | w/Seg | 2.4013 ± 0.0001 | 1.1806 ± 0.0000 | w/Seg | 7.1760 ± 0.0390 | 1.2261 ± 0.0005 |
| TimesNet | 0.4503 ± 0.0006 | 0.5257 ± 0.0004 | TimesNet | 2.7500 ± 0.0004 | 1.2406 ± 0.0001 | TimesNet | | |
| **w/Seg** | **0.2743 ± 0.0001** | **0.3900 ± 0.0001** | w/Seg | 2.3866 ± 0.0012 | 1.1767 ± 0.0002 | w/Seg | — | |
| AGCRN | 0.9951 ± 0.0007 | 0.7359 ± 0.0002 | AGCRN | 2.2722 ± 0.0001 | 1.1458 ± 0.0000 | AGCRN | 7.6546 ± 0.0484 | 1.8930 ± 0.0034 |
| w/Seg | 0.7061 ± 0.0126 | 0.6383 ± 0.0028 | **w/Seg** | **2.0247 ± 0.0049** | **1.0763 ± 0.0007** | w/Seg | 7.4233 ± 0.0472 | 1.8644 ± 0.0034 |
| MTGNN | 0.5926 ± 0.0033 | 0.5539 ± 0.0008 | MTGNN | 2.1919 ± 0.0011 | 1.1175 ± 0.0001 | MTGNN | 7.1258 ± 0.1239 | 1.6903 ± 0.0063 |
| w/Seg | 0.4978 ± 0.0109 | 0.5465 ± 0.0047 | w/Seg | 2.1395 ± 0.0012 | 1.1048 ± 0.0001 | w/Seg | 7.2951 ± 0.0063 | 1.8278 ± 0.0005 |
| GWNet | 0.3953 ± 0.0000 | 0.4573 ± 0.0000 | GWNet | 2.2498 ± 0.0000 | 1.1328 ± 0.0000 | GWNet | 6.8174 ± 0.0616 | 1.5994 ± 0.0039 |
| w/Seg | 0.3732 ± 0.0039 | 0.4691 ± 0.0015 | w/Seg | 2.0856 ± 0.0133 | 1.0981 ± 0.0011 | w/Seg | 7.1551 ± 0.0161 | 1.7891 ± 0.0014 |
| TimeXer | 0.4027 ± 0.0000 | 0.4894 ± 0.0000 | TimeXer | 2.3852 ± 0.0066 | 1.1684 ± 0.0005 | TimeXer | 8.1542 ± 0.7333 | 1.3078 ± 0.0022 |
| w/Seg | 0.3569 ± 0.0002 | 0.4596 ± 0.0001 | w/Seg | 2.2058 ± 0.0022 | 1.1266 ± 0.0002 | w/Seg | 7.3635 ± 0.3326 | 1.2838 ± 0.0026 |
| iTransformer | 0.4772 ± 0.0064 | 0.5004 ± 0.0043 | iTransformer | 2.5433 ± 0.0000 | 1.1808 ± 0.0000 | iTransformer | 8.5086 ± 0.4185 | 1.3474 ± 0.0018 |
| w/Seg | 0.4319 ± 0.0001 | 0.4790 ± 0.0001 | w/Seg | 2.2671 ± 0.0001 | 1.1454 ± 0.0000 | w/Seg | 7.8494 ± 0.0765 | 1.3240 ± 0.0015 |

Table 3: Ablation Study of each module on China Province dataset.

| Method | MSE | MAE | Method | MSE | MAE | Method | MSE | MAE | Method | MSE | MAE | Method | MSE | MAE | Method | MSE | MAE | Method | MSE | MAE |
|---|---|---|---|---|---|---|---|---|---|---|---|---|---|---|---|---|---|---|---|---|
| Informer | 0.8890 | 0.7329 | FEDformer | 0.3588 | 0.4753 | Autoformer | 0.3242 | 0.4490 | TimesNet | 0.4503 | 0.5257 | AGCRN | 0.9951 | 0.7359 | MTGNN | 0.5926 | 0.5539 | GWNet | 0.3953 | 0.4573 |
| **w/ Seg** | **0.7638** | **0.6815** | **w/ Seg** | **0.2941** | **0.3971** | **w/ Seg** | **0.2934** | **0.4008** | **w/ Seg** | **0.2743** | **0.3900** | w/ Seg | 0.7061 | 0.6383 | **w/ Seg** | **0.4978** | **0.5465** | **w/ Seg** | **0.3732** | **0.4691** |
| w/o GE | 0.7805 | 0.6879 | w/o GE | 0.3550 | 0.4574 | w/o GE | 0.3155 | 0.4398 | w/o GE | 0.3461 | 0.4485 | **w/o GE** | **0.7054** | **0.6210** | w/o GE | 0.5357 | 0.5585 | w/o GE | 0.3876 | 0.4822 |
| w/o SE | 0.9188 | 0.7391 | w/o SE | 0.5593 | 0.5526 | w/o SE | 0.3244 | 0.4343 | w/o SE | 0.4333 | 0.5188 | w/o SE | 1.0187 | 0.7452 | w/o SE | 0.5175 | 0.5318 | w/o SE | 0.6250 | 0.5876 |

training dataset. Despite our model achieving the best performance on this dataset, it decreases the performance of two spatio-temporal baselines. This is because the information of these under-represented cities is overwhelmed by nearby larger cities, causing SQGEF to learn representations that include both the obscure city and its surrounding areas. Addressing the noise when modeling obscure geographical entities should be a focus for future work.

## 5.4. Ablation Study(Q2 and Q3)

To delve into the contributions of each module, we perform an ablation study using the China Province dataset. In this study, **w/o SE** denotes models that use only grid embedding solely without hierarchical Segment Quadtree and **w/o GE** denotes models trained exclusively on grid datasets without the Geographical Entity-Based Learning component.

### 5.4.1. SEGMENT QUADTREE EMBEDDING (Q2)

As shown in Table 3, the results show that removing Segment Quadtree Embedding significantly reduces performance. Without this embedding, the model can only capture interactions at the grid level and fails to capture higher-level interactions. This causes the model to perform poorly on province-level data. Spatio-temporal methods AGCRN and GWNet show a significant drop in performance without Segment Quadtree Embedding because they rely on relational graphs. Unlike other spatio-temporal methods like MTGNN, which focus on top-K relations for each entity, AGCRN and GWNet are more affected by noise in the relationships.

### 5.4.2. GEOGRAPHICAL ENTITY-BASED LEARNING (Q3)

As shown in Table 3, removing Geographical Entity-Based Learning decreases performance for all methods. This demonstrates the importance of capturing boundaries in the data. MTGNN shows the largest performance decrease because it uses top-K relations for each entity. Without capturing boundaries, the relationships become too smooth, and the differences between values are smaller. Other methods that include all relationships between entities are less affected, but the top-K approach used by MTGNN can cause the model to miss important relationships.

## 5.5. Parameter Sensitivity(Q4)

To assess the robustness of SQGEF, we evaluate how varying parameter choices influence the model's performance. All parameters are set to their default values except for the one being tested. The embedding size denotes the number of dimensions for each node in the segment quadtree, while the temperature factor regulates the smoothness of the constructed graph in the softmax function. Our findings indicate that the method is not sensitive to minor variations in these parameters across all three experiments. It maintains high performance with cost-effective parameter settings, demonstrating both its effectiveness and robustness. Specifically, an embedding size of 18 yields optimal performance for the China Province and China City datasets, whereas a size of 20 is more suitable for the Europe Country dataset. This suggests that the Europe Country dataset exhibits greater variability and thus requires a larger embedding size to cap-

ture it effectively. For the temperature factor, the best performance is observed at 0.95 for the China Province dataset, which differs from the optimal settings for the other two datasets. This implies the China Province dataset contains many similar provinces, necessitating a lower temperature factor to accurately differentiate between them.

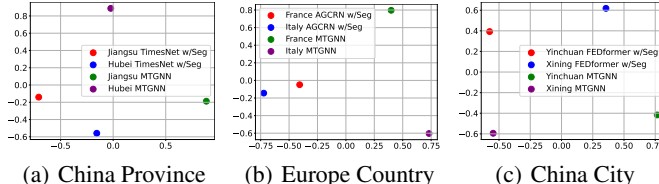

(a) China Province  (b) Europe Country  (c) China City

Figure 5: Embeddings learned from Segment Quadtree compared to MTGNN on all datasets

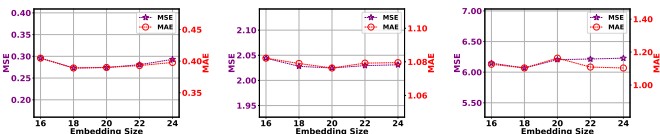

(a) TimesNet w/Seg on China Province  (b) AGCRN w/Seg on Europe Country  (c) FEDformer w/Seg on China City

Figure 3: Parameter sensitivity of Segment Quadtree with Embedding Size in terms of MSE and MAE

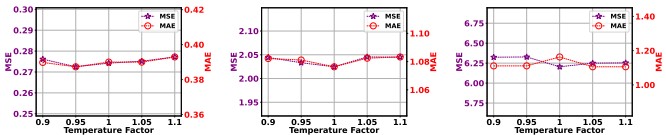

(a) TimesNet w/Seg on China Province  (b) AGCRN w/Seg on Europe Country  (c) FEDformer w/Seg on China City

Figure 4: Parameter sensitivity of Segment Quadtree with Temperature Factor in terms of MSE and MAE

### 5.6. Case Study(Q5)

In this section, we compare the quality of embeddings generated by our method with those that adaptively construct the relation graph from the downstream dataset, such as MTGNN. The entities selected for comparison are based on their similarity in economic or industrial indicators, ensuring a meaningful evaluation of embedding quality. The results highlight our method's superior ability to represent geographical entities. First, we compare the embeddings of Jiangsu and Hubei. Both provinces exhibit similar carbon emission patterns due to their high concentration of energy-intensive industries. Figure 5(a) shows that embeddings learned from our method are much closer to each other compared to those from MTGNN. Similarly, Figure 5(b) compares the embeddings of France and Italy. Both countries are notable for their positive efforts in expanding renewable energy and improving energy efficiency, setting them apart from other EU countries. Our method yields embeddings that are closer to each other than those produced by MTGNN. Finally, we compare the embeddings of Yichuan and Xining, two inland, semi-arid cities in China that rely heavily on fossil fuels. Our method provides embeddings that are closer to each other than those from MTGNN. These experiments demonstrate that our method achieves better representations of geographical entities compared to the embeddings adaptively learned from the downstream dataset.

### 5.7. Cross-Domain Generalization Performance(Q6)

To further demonstrate our framework's adaptability across domains, we conducted additional experiments using air pollution datasets. The results confirm the model's versatility and effectiveness across different domains, with detailed results and settings provided in the appendix.

## 6. Related Work

Our related work can be summarized into two categories: Spatio-temporal forecasting and Scientific computing .

### 6.1. Spatio-Temporal Forecasting

Spatio-temporal forecasting has applications across various granularity geographical granularities. At the country level, Jiao *et al.* leverage spatial dependence and heterogeneity to predict tourist arrivals across 37 European countries (Jiao et al., 2020). At the provincial or state level, earlier studies have examined the demand for inbound tourism within different provinces of a single country (Yang & Zhang, 2019). At the city level, previous studies examined the composition and spatio-temporal variations of municipal solid waste (Mushtaq et al., 2020) and job mobility (Zhang et al., 2021). At the street level, RDAT (Liu et al., 2023a) focuses on making Intelligent Transportation Systems resilient to adversarial attacks, while MM-DAG (Lan et al., 2023) introduces a multi-task learning algorithm for managing multi-modal traffic at intersections. It is common to utilize graphs to model the relationships between entities, thereby enhancing forecasting. The graph extracts relevant information from similar nodes, improving the accuracy (Liu et al., 2022b) and robustness (Li et al., 2022; Liu et al., 2023d; Li et al., 2024) of predictions. However, these methods primarily focus on utilizing historical data of forecasting targets, overlooking the potential of other heterogeneous datasets. To address this, we propose Segment Quadtree Embedding to extract information from heterogeneous datasets with different granularities and data structures, providing a comprehensive representation of entities within a region.

### 6.2. Scientific Computing

Scientific computing leverages advanced computational capabilities to analyze complex natural problems. In the cli-

mate field, understanding the correlation between different geographical entities is crucial for effective environmental strategies, such as forecasting carbon emissions (Peters et al., 2012) and earthquake (Zhu et al., 2020). Astsatryan *et al.* utilize deep learning models to predict city temperatures for weather forecasting (Astsatryan et al., 2021). In geology, Pao *et al.* employ finite element methods (FEM) to simulate the complex interactions across regions to forecast earthquakes (Paolucci et al., 2018), while Taborda *et al.* use numerical methods to predict earthquake effects in urban areas to aid urban planning (Taborda & Roten, 2015). In social science, AI techniques are used to detect financial fraud (Liu et al., 2021; Huang et al., 2022; Gong & Sun, 2024) and model recruitment variations (Sun et al., 2021; 2024; Qin et al., 2025b). In biology, AI is used to identify protein and gene functions (Bao & Yang, 2024; Wang et al., 2024). Furthermore, AI is increasingly applied to solve scientific problems (Qin et al., 2025a), including mathematical problem-solving (Liu et al., 2023c;b; 2022a; Lin et al., 2021) and causal attribution analysis (Ji et al., 2025). However, scientific computing often faces data scarcity. The high cost of collecting comprehensive data limits analyses to task-specific datasets, resulting in scientific datasets with few observation points and short time spans, making it difficult to provide sufficient historical information for model training. To address this, we propose SQGEF to unify knowledge from diverse datasets, enhancing forecasting on target datasets.

### 6.3. Quadtree Applications in Spatial Data Management

In the domain of spatial data management, Kothuri et al. in their study (Kothuri et al., 2002) investigate the application of quadtree and R-tree indexes within Oracle Spatial, comparing their performance in query efficiency and storage using geographic information system (GIS) data. Conversely, Yin et al. in their work (Yin et al., 2011) focus on the role of quadtrees in the representation and compression of spatial data, proposing a method to optimize data storage and processing efficiency using quadtree structures.

## 7. Conclusion

In this paper, we introduce the Segment Quadtree Geographical Embedding Framework (SQGEF), a novel approach designed to integrate diverse heterogeneous datasets and address data scarcity in scientific research. SQGEF employs a novel data structure, the Segment Quadtree, which hierarchically represents entities of varying granularities and accommodates previously unseen entities. Additionally, we design learning methods for both grid and geographical entity datasets, capturing multi-level interactions and geographical entity knowledge, such as nested relationships and human-defined boundaries from diverse entities, enabling a comprehensive understanding of complex geographical

structures. Analyses and experiments with datasets from various regions and granularities demonstrate that our method effectively represents different unseen geographical entities and improves the performance of various models. By integrating heterogeneous datasets and providing relationships for unseen entities, SQGEF offers valuable support for emerging scientific tasks facing data scarcity.

## Impact Statement

This work aims to enhance spatio-temporal forecasting for geographical scientific data by integrating heterogeneous datasets. Our approach has potential applications in environmental monitoring and resource management. There are many potential societal consequences of our work, none of which we feel must be specifically highlighted here.

## Acknowledgements

This work is partly supported by the National Natural Science Foundation of China (No. 62306255, 92370204), the National Key Research and Development Program of China (No. 2023YFF0725000), the Guangdong Basic and Applied Basic Research Foundation (No. 2024A1515011839), the Fundamental Research Project of Guangzhou (No. 2024A04J4233), and the Education Bureau of Guangzhou Municipality.

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

# A. Technical Appendix

## A.1. Data Description

ODIAC

The ODIAC (oda, 2011) provides high-resolution global data on fossil fuel CO2 emissions. It combines country-level estimates from the Carbon Dioxide Information Analysis Center (CDIAC) with spatial proxies such as point source locations and satellite nightlight data, covering the years 2000-2021. We employ two different resolutions: 1km $\times$ 1km and 1deg $\times$ 1deg grid subsets at China and Europe locations.

MEIC

The MEIC (Tong et al., 2020; Cheng et al., 2021; 2023) is a platform to model atmospheric emissions from human activities. It generates detailed emission inventories for China, supporting scientific research and policy evaluation from 1997 to 2017. The region subset is regional-scale, split by industrial zones and cities, including some obscure cities.

CARBON MONITOR

Carbon Monitor (Liu et al., 2020) offers daily, up-to-date estimates of CO2 emissions from fossil fuels and cement production. This international initiative tracks crucial data for climate change studies from 2019 to 2023. We choose the China and Europe regions for our experiments.

## A.2. Baselines

**Informer** (Zhou et al., 2021) proposes the ProbSparse self-attention mechanism to replace inner product self-attention, thereby reducing time and space complexity.

**FEDformer** (Zhou et al., 2022) designs two attention modules that utilize the Fourier transform and wavelet transform to perform the attention operation in the frequency domain.

**Autoformer** (Wu et al., 2021) is a transformer-style model that introduces the Auto-Correlation Mechanism to replace the dot-product attention mechanism. It also presents the Decomposition Architecture, which extracts more predictable components from complex temporal patterns.

**TimesNet** (Wu et al., 2022) models two types of temporal changes: within a modeling period (Intraperiod) and between periods (Interperiod). It expands one-dimensional temporal data into a two-dimensional space for analysis.

**AGCRN** (Bai et al., 2020) combines graph convolutional networks with recurrent neural networks to capture both spatial and temporal dependencies in time series data.

**MTGNN** (Wu et al., 2020) extracts relationships among variables using graph learning module. It integrates external knowledge and captures spatial and temporal dependencies from the data.

**GWNet** (Wu et al., 2019a) integrates graph convolutional networks with WaveNet architecture to effectively model spatial-temporal correlations in sequential data.

## A.3. Implementation Details

We implemented our model using PyTorch 2.1.0, based on Python 3.8.18, and utilized a Tesla A6000 GPU for training. The dimensions of node embeddings are set to 20. We used the Adam optimizer (Kingma & Ba, 2015) for parameter optimization. The learning rate was set to 0.0001 and the dropout rate to 0.1. We employ Gaussian initialization for the node embeddings to provide a robust starting point for training. The graph is constructed by calculating the interaction strengths between nodes through the product of their embeddings. Specifically, we compute the pairwise dot product of node embeddings to form the adjacency matrix. This is followed by softmax normalization with a temperature parameter applied to each row of the adjacency matrix to control the sharpness of the distribution, resulting in the final interaction probabilities. For each experiment, we trained the model in the order listed in Table 1.

## A.4. Cross-Domain Generalization Performance

To better demonstrate our framework's adaptability across various domains, we have included additional experiments using air pollution datasets. We trained our model with PM2.5 and PM10 grid datasets and tested it using O3 data at the provincial level. The results affirm the model's versatility and effectiveness across different domains.

| Method | MSE | MAE |
|---|---|---|
| Informer | 2.0194 | 1.1581 |
| w/Seg | 1.9953 | 1.1510 |
| FEDformer | 2.0272 | 1.1584 |
| w/Seg | 1.7034 | 1.0619 |
| Autoformer | 2.1276 | 1.1355 |
| w/Seg | 2.0754 | 1.1215 |
| AGCRN | 2.7700 | 1.3346 |
| w/Seg | 2.2893 | 1.2133 |
| MTGNN | 2.5206 | 1.2658 |
| w/Seg | 2.3239 | 1.2139 |
| GWNet | 2.5746 | 1.2803 |
| w/Seg | 2.5782 | 1.2796 |

Table 4: Performance on air pollution datasets.

## A.5. Detailed Proof of Generalization Bound for SQGEF

We prove that the SQGEF yields a tighter generalization error bound compared to the traditional Grid-based Partitioning Embedding using Rademacher complexity.

### A.5.1. PROBLEM SETUP AND NOTATION

Let $S = \{(x_1, y_1), \ldots, (x_n, y_n)\}$ be a training set of $n$ samples drawn i.i.d. from an unknown distribution $\mathcal{D}$ over $\mathcal{X} \times \mathcal{Y}$. For a hypothesis $h \in \mathcal{H}$, define the true error as $L(h) = \mathbb{E}_{(x,y)\sim\mathcal{D}}[\ell(h(x), y)]$ and the empirical error as $\hat{L}_S(h) = \frac{1}{n}\sum_{i=1}^{n}\ell(h(x_i), y_i)$, where $\ell : \mathcal{Y} \times \mathcal{Y} \to [0, 1]$ is a bounded loss function.

We compare two embedding methods for an entity with $k$ grids region: **Grid-based Partitioning Embedding**: Fuses $k$ grid embeddings using $k$ embeddings. The hypothesis class is denoted $\mathcal{H}_{\text{grid}}$. **Segment Quadtree Embedding**: Constructs a segment tree over the grids of region, fusing $m = O(\log k)$ node embeddings to represent the entity. The hypothesis class is denoted $\mathcal{H}_{\text{seg}}$.

### A.5.2. GENERALIZATION BOUND WITH RADEMACHER COMPLEXITY

We rely on a standard result from statistical learning theory:

**Theorem A.1** (Rademacher Complexity Bound (Bartlett & Mendelson, 2002))**.** *For any hypothesis class $\mathcal{H}$, with probability at least $1 - \delta$ over the draw of S, for all $h \in \mathcal{H}$,*

$$L(h) \leq \hat{L}_S(h) + 2\hat{\mathcal{R}}_S(\mathcal{H}) + 3\sqrt{\frac{\ln(2/\delta)}{2n}},$$

*where $\hat{\mathcal{R}}_S(\mathcal{H}) = \frac{1}{n}\mathbb{E}_\sigma\left[\sup_{h\in\mathcal{H}}\sum_{i=1}^{n}\sigma_i\ell(h(x_i), y_i)\right]$ is the empirical Rademacher complexity, and $\sigma_i \sim \text{Unif}(\{+1, -1\})$ are Rademacher variables.*

The generalization gap $L(h) - \hat{L}_S(h)$ is controlled by $\hat{\mathcal{R}}_S(\mathcal{H})$, which measures the complexity of $\mathcal{H}$. We now bound $\hat{\mathcal{R}}_S(\mathcal{H}_{\text{grid}})$ and $\hat{\mathcal{R}}_S(\mathcal{H}_{\text{seg}})$.

### A.5.3. COMPLEXITY ANALYSIS

For clarity, we consider an MLP to fuse $k$ embeddings without loss of generality. The Grid-based Partitioning Embedding method employs an MLP to map $k$ embeddings, each in $\mathbb{R}^d$, into a single $d$-dimensional embedding. The input is a

concatenation of $k$ vectors (dimension $kd$), with $L$ hidden layers and parameter count $p$ scaling with $kd$. The Rademacher complexity is bounded as (Bartlett et al., 2017):

$$\hat{\mathcal{R}}_S(\mathcal{H}_{\text{grid}}) \leq C_1 \cdot \frac{\|W\|^{L+1}\sqrt{p}\sqrt{kd}}{\sqrt{n}},$$

where $C_1 > 0$ is a constant, $\|W\|$ is the maximum spectral norm of weight matrices, and $p \propto kd$ due to input layer weights.

Similarly, the Segment Quadtree Embedding fuses $m = O(\log k)$ embeddings, each in $\mathbb{R}^d$, into a single $d$-dimensional embedding using an MLP. The input dimension is $md$, with $L$ hidden layers and parameter count $p'$ scaling with $md$. The Rademacher complexity is bounded as:

$$\hat{\mathcal{R}}_S(\mathcal{H}_{\text{seg}}) \leq C_2 \cdot \frac{\|W\|^{L+1}\sqrt{p'}\sqrt{md}}{\sqrt{n}},$$

where $C_2 > 0$ is a constant, and $p' \propto md$.

### A.5.4. COMPARISON OF BOUNDS

Assume $L$ and $|W| > 1$ are the same for both methods, and the constants $C_1 \approx C_2$ are approximately equal, with $p \propto kd$ and $p' \propto md$. The Rademacher complexities are $\hat{\mathcal{R}}_S(\mathcal{H}_{\text{grid}}) \propto \|W\|^{L+1}\sqrt{kd}\sqrt{p}$ for Grid-based Partitioning Embedding and $\hat{\mathcal{R}}_S(\mathcal{H}_{\text{seg}}) \propto \|W\|^{L+1}\sqrt{md}\sqrt{p'}$ for Segment Quadtree Embedding. Since $m = O(\log k) < k$, and $p' < p$, we have $\sqrt{md}\sqrt{p'} < \sqrt{kd}\sqrt{p}$, thus $\hat{\mathcal{R}}_S(\mathcal{H}_{\text{seg}}) < \hat{\mathcal{R}}_S(\mathcal{H}_{\text{grid}})$.

Applying Theorem A.1, for $h_{\text{grid}} \in \mathcal{H}_{\text{grid}}$,

$$L(h_{\text{grid}}) \leq \hat{L}_S(h_{\text{grid}}) + 2\hat{\mathcal{R}}_S(\mathcal{H}_{\text{grid}}) + 3\sqrt{\frac{\ln(2/\delta)}{2n}}.$$

For $h_{\text{seg}} \in \mathcal{H}_{\text{seg}}$,

$$L(h_{\text{seg}}) \leq \hat{L}_S(h_{\text{seg}}) + 2\hat{\mathcal{R}}_S(\mathcal{H}_{\text{seg}}) + 3\sqrt{\frac{\ln(2/\delta)}{2n}}.$$

Given that $\hat{\mathcal{R}}_S(\mathcal{H}_{\text{seg}}) < \hat{\mathcal{R}}_S(\mathcal{H}_{\text{grid}})$, the generalization bound for the Segment Quadtree Embedding is tighter.

### A.6. Data Scarcity

Data scarcity primarily refers to the fact that all forecasting targets in the test set are **unseen** during pretraining, meaning their historical data is entirely absent from the pretraining stage. Consequently, the data available to directly model relationships between these targets is extremely limited. To address this, SQGEF leverages a large volume of heterogeneous datasets from related regions or entities, even though these datasets do not contain the historical records of test targets. For example, in the China Province experiment, we aim to forecast future carbon emissions for several Chinese provinces. During pretraining, we utilize two distinct grid-based datasets and historical records of various regions, but none of these include the historical data of the target provinces themselves. This approach allows us to infer patterns indirectly.

To quantify this scarcity, we compare the data volume between the pretraining and test stages across three metrics: total data points, number of temporal points, and temporal duration. The table below summarizes these statistics for our three experimental settings:

From this table, it is evident that the test set contains significantly fewer data points than the pretraining set across all experiments. For China Province and Europe Country, the test set's temporal duration is notably short (2 years). In the China City, test set has an extremely limited number of temporal records (37 points), indicating sparse sampling. Collectively, these statistics highlight the severe data scarcity in the test set, both in terms of quantity and coverage.

### A.7. Complexity Analysis

**Space Complexity**: Both SQGEF and the naive grid embedding method have a space complexity of $O(N)$ (where $N$ is the number of grid cells). SQGEF introduces only a constant-level overhead to store multi-level interaction information.

Table 5: Data scarcity comparison between pretraining and test sets

| Experiment | Pretrain Set Data Points | Test Set Data Points | Pretrain Set Time Points (Span) | Test Set Time Points (Span) |
|---|---|---|---|---|
| China Province | 8,651,704 | 37,062 | 581 (22 years) | 1,278 (2 years) |
| Europe Country | 8,650,752 | 39,618 | 528 (22 years) | 1,278 (2 years) |
| China City | 8,687,814 | 962 | 1,806 (22 years) | 37 (37 years) |

In our Segment Quadtree, the total number of nodes is $N$ (root level) $+ (1/4)N$ (next level) $+ \ldots + (1/4)^h N$, where $h = \lceil \log_4 N \rceil$ is the tree height. This geometric series sums to $O((4/3)N)$, which remains linear in $N$. Thus, our approach scales efficiently even with higher-resolution grids, incurring minimal additional storage costs.

**Training Stage Time Complexity**: While both SQGEF and the naive grid embedding method have a computational complexity of $O(N^2)$ (where $N$ is the number of grid cells), SQGEF adds only a constant overhead to process multi-granularity data. While SQGEF models interactions across all granularity levels, the additional computational cost is minor relative to the substantial performance gains it delivers. The complexity comprises two parts: (1) constructing a graph to capture spatial relations between grids, and (2) computing temporal relations across timestamps. For the graph, the naive method's complexity is $O(N^2)$. In SQGEF, this becomes $O(N^2 + (1/16)N^2 + \cdots + (1/16)^h N^2)$ across tree levels, summing to $O((16/15)N^2)$, a slight increase over $O(N^2)$. Temporal relations are treated as a constant $T$, as no additional temporal computation is required beyond the input sequence. This modest overhead enables rich multi-level interactions, significantly enhancing forecasting accuracy, as evidenced by our experimental results.

**Inference Stage Time Complexity**: During inference, SQGEF outperforms the naive grid embedding method in efficiency. Using the segment tree query method [1] for a region $[qX_1, qX_2] \times [qY_1, qY_2]$, we recursively check overlaps with the current node's region $[X_1, X_2] \times [Y_1, Y_2]$. With a tree height of $O(\log_4 N)$, and up to 4 nodes visited per level (if the query spans quadrants), the total nodes visited is bounded by $4 \cdot O(\log_4 N) = O(\log N)$ (since $\log_4 N = (1/2) \log_2 N$). This contrasts with the naive grid method's $O(N)$ complexity, making our approach far more efficient.

**Summary**: During training, SQGEF maintains $O(N)$ space complexity and $O(N^2)$ computational complexity, identical to the naive method, enabling the capture of multi-granularity relationships that yield significant performance improvements. In inference, our method achieves $O(\log N)$ complexity, versus $O(N)$ for the naive approach, ensuring superior scalability and efficiency. These trade-offs make SQGEF highly practical and advantageous, especially for numerous downstream spatio-temporal forecasting tasks, where its fast inference speed delivers substantial benefits across a wide range of applications.

