# OpenReview forum: "Unifying Knowledge from Diverse Datasets to Enhance Spatial-Temporal Modeling: A Granularity-Adaptive Geographical Embedding Approach"
_ICML.cc/2025/Conference — ICML 2025 poster_

### Official Review · Reviewer_1nAg · 2025-03-07

**Overall Recommendation:** 3

**Summary:**

Spatio-temporal datasets are often heterogeneous, each with its own granularity and representation of entities. Given the data scarcity issue in such datasets, it is crucial to integrate them effectively and learn unified representations for multi-granularity entities. The authors propose a framework to achieve this, called the Segment Quadtree Geographical Embedding Framework (SQGEF). Their approach leverages a quadtree data structure to embed entities of varying granularities. They evaluate SQGEF on three different datasets for spatio-temporal forecasting of carbon emissions and demonstrate that their method generally improves baseline performance for this task.

**Claims And Evidence:**

The approach proposed by the authors is particularly useful for spatio-temporal datasets and could be applied to various applications. Their method appears to improve existing approaches for spatio-temporal forecasting. However, the authors claim to "propose a novel data structure" for spatial representation—specifically, segment quadtrees (see abstract). Quadtrees have long been used for spatial data representation, as evidenced by works such as Kothuri et al. (2002). Yet, the authors do not reference any prior work related to quadtrees.
I would like to understand the rationale behind omitting such an important body of work. Additionally, what distinguishes a segment quadtree from a standard quadtree? I believe the authors should clarify this distinction in the text.

Kothuri, Ravi Kanth V., Siva Ravada, and Daniel Abugov. "Quadtree and R-tree indexes in oracle spatial: a comparison using GIS data." In Proceedings of the 2002 ACM SIGMOD international conference on Management of data, pp. 546-557. 2002.

**Essential References Not Discussed:**

References to quadtrees for spatial representations are missing, such as:

Kothuri, Ravi Kanth V., Siva Ravada, and Daniel Abugov. "Quadtree and R-tree indexes in oracle spatial: a comparison using GIS data." In Proceedings of the 2002 ACM SIGMOD international conference on Management of data, pp. 546-557. 2002.

Yin, Xiang, Ivo Düntsch, and Günther Gediga. Quadtree representation and compression of spatial data. Springer Berlin Heidelberg, 2011.

**Experimental Designs Or Analyses:**

I believe Figure 5 is misleading and cannot be interpreted as the authors suggest. What do the two axes represent? Did the authors perform dimensionality reduction on the embedding space? If so, this should be clearly explained. Moreover, if dimensionality reduction was applied, I do not think it is appropriate to compare the embedding spaces of two models in this way, as distances between points, especially after reduction, may not be directly comparable. There should be more than two points per model to begin to understand the structure of the embedding space.

**Methods And Evaluation Criteria:**

The method performs well on the three proposed downstream tasks. However, in my opinion, the experimental datasets are somewhat limited for evaluating this framework for two main reasons:
- It focuses on a single spatio-temporal application—carbon emissions.
- The three tasks are closely related, with two of them differing only by an inversion of the training and testing datasets, and the other two using different regions.

Ideally, I would like to see an additional experiment with a different setup to better assess whether the method can be generally applied to various spatio-temporal modeling scenarios, as suggested by the title.

Additionally, the datasets and their spatial distribution are barely discussed in the main text and are only briefly covered in the supplementary materials.

Edit: After reading the supplementary material, I noticed the experiment on the air pollution dataset, which is not mentioned in the main text. I believe this experiment should be given more emphasis and better described, including relevant references.

**Other Comments Or Suggestions:**

Line 107: $N^{test}$ instead of $N$

**Other Strengths And Weaknesses:**

I believe the paper lacks clarity due to missing details. For example, in Equation (1), $e_{k,l}$ are introduced for the first time without any explanation of what they are and how they are obtained. Later, they are referred to as embeddings, but it seems to be never explained how they are derived or trained.

However, the method may have significant value, and the setup considered is interesting to explore. It could also benefit many other applications.

**Questions For Authors:**

I would like the authors to address the following points raised earlier:
- Clarification and references to quadtrees
- Lack of diversity in datasets considered and a more detailed dataset description
- Improvement to the theoretical analysis section
- Addressing potential issues with Figure 5
- More references
- General clarification of the text

I am willing to increase my recommendation if the authors address these points, as the method has potential.

**Relation To Broader Scientific Literature:**

As mentioned earlier, there is a lack of references and connections to previous works on using quadtrees for spatial representations. Additionally, the paper has a limited number of references (only 24), with no citations from the location encoding field.

**Theoretical Claims:**

I find the theoretical analysis unclear. For example, it is not clear for me where the generalization error bound comes from or what the terms $C$ and $n$ represent. I believe more details should be provided, either through additional references or in the appendix.

---

> ### Author Rebuttal · Authors · 2025-04-01
>
> Thank you for your detailed and thorough feedback on our work. We greatly appreciate your insightful comments. Due to space constraints, we are unable to incorporate all of this content into the main text. Since URLs are not permitted (except for figures), we have instead included an updated PDF with detailed proofs in the supplementary materials, which has been uploaded to an anonymous repository. All referenced appendices are provided in full. We sincerely hope you will review these additional materials and reconsider our score.
>
> # W1
> The idea of an indextree in references you given is to efficiently index spatial entities by hierarchically partitioning a region into proper granularity levels but coarser than entity level (containing multiple cities), implemented as a quadtree where child nodes separate the region of parent nodes, bring no additional information. Our work, neither motivated by nor following indextree, addresses the problem of using heterogeneous datasets to extrapolate knowledge to model unseen entities, a challenge indextree cannot solve due to its focus on summarizing known entities without extrapolation capabilities. To tackle this, we use finer granularity nodes (grids below city level), stores distinct information across parent-child levels from different granularity datasets (detailed discussion in Appendix A.9), and employs a segmenttree query method to fuse mixed-level embeddings, capturing different-level interactions. In summary, our Segment Quadtree is a new data structure designed for knowledge extrapolation, distinct from indextree’s indexing paradigm.
>
> We listed more detailed foundational differences between indextree and Segment Quadtree are shown in Appendix A.7.
>
> The "segment" in Segment Quadtree reflects its query method, inspired by segment trees, a data structure for efficiently computing range sums over continuous arrays. For example, in a segment tree with values (a1,a2, a3,a4), querying the sum of (a1, a2,a3) uses precomputed nodes like ([a1, a2]) and (a3), reducing computation. Similarly, in our Segment Quadtree in Figure 2(a), querying an yellow entity’s boundary fuses embeddings from mixed levels: one high-level node (Node3), one mid-level node (Node17), and two low-level nodes (Nodes69,71), rather than aggregating all lowest-level grids, as done for France in Figure 1(b). This multi-level fusion accelerates queries and leverages hierarchical information, distinguishing it from standard quadtrees like indextrees, where fusing across levels is less meaningful due to uniform node content.
>
>
> # W2
> Due to constraints in a paper, we cannot conduct experiments across a wider range of scenarios. Therefore, we focused on two key cases to allow for a more thorough analysis. We acknowledge this limitation and will explore additional applications of our method in future work.
>
> To address the air pollution datasets description concern, we have improved the dataset description for air pollution in Appendix A.1.
>
> # W3
> We have added a comprehensive proof in Appendix A.5. Due to space constraints, we only summarize the key conclusions here.
> From the Rademacher Complexity Theorem, for any $\mathcal{H}$, the generalization error bound is $L(h) \leq L^S(h) + 2\hat{\mathcal{R}}_S(\mathcal{H}) + 3\sqrt{\frac{2\ln(2/\delta)}{n}}$, where $\hat{\mathcal{R}}_S(\mathcal{H})$ is the empirical Rademacher complexity. For Segment Quadtree Embedding, $\hat{\mathcal{R}}S(\mathcal{H}{\text{seg}}) \leq C_2 \cdot \frac{|W|^{L+1} \sqrt{p'} \sqrt{m d}}{\sqrt{n}}$, with $m = O(\log k)$ and $p' \propto m d$, giving a tighter bound for Segment Quadtree Embedding.
>
> # W4
> Following your recommendation, we shifted from visualization-based analysis to a quantitative evaluation of embedding distances on all embeddings. Specifically, we computed distances between all pairs of embeddings and compared them with the corresponding time series distances of carbon emission amounts. For the carbon emission time series, we calculated the L2 norm between every pair of provinces. Similarly, for each model, we computed the L2 distances between all pairs of embeddings. We then assessed the correlation between these embedding distances and the real carbon emission distances using the Pearson correlation coefficient. The results demonstrate that the Pearson score for TimesNet w/Seg embeddings is significantly higher than that of MTGNN.
>
> |Exp|TimesNet w/Seg|MTGNN|
> |-|-|-|
> |China Province|0.377|0.121|
> |Europe Country|0.480|0.343|
> |China City|0.247|0.226|
> Results indicate that our method better captures the structure of entities compared to MTGNN.
>
> # W5
> We have written an related work section about Quadtree Applications in Spatial Data Management in Section 6.3.
>
> # W6
> We have clarified the differences between the Index Tree and Segment Quadtree, added air pollution experiments, provided a detailed theoretical and complexity analysis, and quantative data scarcity in the original paper.

---

> > ### Comment · Reviewer_1nAg · 2025-04-03
> >
> > I appreciate the additional efforts the authors have made to address my concerns, particularly the development of the theoretical analysis, the clarification and references to quadtrees, and fixing the analysis of embedding distances. I hope some of these changes will be included in the main text. Ideally, I would have liked to see experiments across a wider range of scenarios, but I understand the constraints of this field and how difficult it is to incorporate other applications. Accordingly, I will increase my recommendation.

---

> > > ### Author Response · Authors · 2025-04-03
> > >
> > > Thank you so much for the improvement in the score. Your valuable suggestions and the additional supplementary references to prior work have made our paper more solid and significantly improved the clarity of our contributions. We will incorporate them into the main text to further strengthen the paper, following your suggestions, and explore their potential applications in future work.
> > >
> > > Thank you again for your time and effort in providing such a thorough review! Your feedback has been truly invaluable in improving our work.

---

### Official Review · Reviewer_y3tt · 2025-03-14

**Overall Recommendation:** 3

**Summary:**

This paper addresses the challenge of data scarcity in geographical scientific datasets for spatio-temporal forecasting by proposing a novel framework called Segment Quadtree Geographical Embedding Framework (SQGEF). The framework integrates knowledge from diverse datasets with varying granularities, time spans, and observation variables to learn unified representations for multi-granularity entities, including those absent during training. The key contributions include a novel method for integrating heterogeneous datasets, a unique embedding approach for different granularity entities, and comprehensive experiments showcasing the method's effectiveness and robustness across various regions and granularities.

## Update after rebuttal: Thanks for author rebuttal and I have read the rebuttal. I think my concerns are addressed and will keep my score this time (for weak acceptance).

**Claims And Evidence:**

The claims are supported by experimental result.

**Essential References Not Discussed:**

The coverage of references is good.

**Experimental Designs Or Analyses:**

The overall designs of experiment are acceptable.

**Methods And Evaluation Criteria:**

The proposed method is technically sound.

**Other Comments Or Suggestions:**

See above.

**Other Strengths And Weaknesses:**

**Strengths:**
- Novel and interesting idea. The Segment Quadtree data structure provides a hierarchical representation of geographical regions, capturing multi-level interactions and geographical entity knowledge
- The method shows improvements across various models, including time series and spatio-temporal models, and has potential applications in different scientific domains.

**Weaknesses:**
- Authors should provide the information about data scarcity of utilized datasets.
- It will be better to compare the performance gains from combining data of different granularity compared to using a single data set.

**Questions For Authors:**

See above.

**Relation To Broader Scientific Literature:**

The Segment Quadtree is novel in the literature. I appreciate the interesting idea.

**Theoretical Claims:**

The authors try to give theoretical analysis of the method, but more details are needed to make it clear and convincing, e.g., the calculation of bounds should be provided.

---

> ### Author Rebuttal · Authors · 2025-04-01
>
> Thank you for your valuable feedback on our work. We greatly appreciate your insightful comments, including your detailed suggestions regarding the calculation of bounds, clarifications on data scarcity, and additional experiments. Due to space constraints, we are unable to incorporate all of this content into the main text. Since URLs are not permitted (except for figures), we have instead included an updated PDF with detailed proofs in the supplementary materials, which has been uploaded to an anonymous repository. All referenced appendices are provided in full. We sincerely hope you will review these additional materials and reconsider our score.
>
> # the calculation of bounds
>
> We have added a comprehensive proof in Appendix A.5. Due to space constraints, we only summarize the key conclusions here.
> From the Rademacher Complexity Theorem, for any $\mathcal{H}$, the generalization error bound is $L(h) \leq L^S(h) + 2\hat{\mathcal{R}}_S(\mathcal{H}) + 3\sqrt{\frac{2\ln(2/\delta)}{n}}$, where $\hat{\mathcal{R}}_S(\mathcal{H})$ is the empirical Rademacher complexity. For Segment Quadtree Embedding, $\hat{\mathcal{R}}S(\mathcal{H}{\text{seg}}) \leq C_2 \cdot \frac{|W|^{L+1} \sqrt{p'} \sqrt{m d}}{\sqrt{n}}$, with $m = O(\log k)$ and $p' \propto m d$, giving a tighter bound for Segment Quadtree Embedding.
>
> # W1
>
> Thank you for your valuable suggestion! We first clarify what data scarcity means in our work and then provide statistical evidence to support it. Due to space constraints, we have added detailed description in Appendix A.6.
>
> In our paper, data scarcity refers to forecasting targets in the test set being unseen during pretraining, with no historical data available at that stage. This severely limits direct modeling of their relationships. To tackle this, SQGEF uses a large volume of heterogeneous datasets from related regions or entities, despite lacking test target historical records.To quantify this scarcity, we compare the data volume between the pretraining and test stages across three metrics: total data points, number of temporal points, and temporal duration.
>
> |Experiment|Pretrain Set Data Points|Test Set Data Points|Pretrain Set Time Points (Span)|Test Set Time Points (Span)|
> |-|-|-|-|-|
> |China Province|8,651,704|37,062|581 (22 years)|1,278 (2 years)|
> |Europe Country|8,650,752|39,618|528 (22 years)|1,278 (2 years)|
> |China City|8,687,814|962|1,806 (22 years)|37 (37 years)|
>
> From this table, it is evident that the test set contains significantly fewer data points than the pretraining set across all experiments. For China Province and Europe Country, the test set’s temporal duration is notably short(2 years). In the China City, test set has an extremely limited number of temporal records(37 points), indicating sparse sampling. Collectively, these statistics highlight the severe data scarcity in the test set, both in terms of quantity and coverage.
>
> Moreover, Table 2 in our paper shows that this scarcity goes beyond data volume to informational richness. Models using only the scarce historical data of forecasting targets consistently lag behind SQGEF, which combines this limited target data with richer, more plentiful data from related grids and entities during pretraining.
>
> # W2
>
> Thank you for your suggestion! This will help clarify the effect of different granularity datasets. We use only one grid dataset to train the model. As shown in the following table, using one single dataset significantly drops the performance, which verifies the validity of unifying various datasets.
>
> ||MSE|MAE|
> |-|-|-|
> |Informer|0.7983|0.7036|
> |FEDformer|0.3628|0.4674|
> |Autoformer|0.3241|0.4518|
> |TimesNet|0.3985|0.5162|
> |AGCRN|0.8112|0.7142|
> |MTGNN|0.5518|0.5753|
> |GWNet|0.3984|0.4954|

---

> > ### Comment · Reviewer_y3tt · 2025-04-03
> >
> > Thanks for author rebuttal and  I will maintain my score at this time.

---

> > > ### Author Response · Authors · 2025-04-03
> > >
> > > Thank you for your constructive feedback and for giving us the opportunity to improve our manuscript. Your insightful comments regarding the data scarcity discussion and additional experiments have significantly strengthened our paper.
> > >
> > > We sincerely appreciate the time and effort you dedicated to reviewing our work.

---

### Official Review · Reviewer_pADa · 2025-03-14

**Overall Recommendation:** 3

**Summary:**

This paper tackles the challenge of data scarcity in geographical scientific datasets for spatio-temporal forecasting by introducing a novel framework: the Segment Quadtree Geographical Embedding Framework (SQGEF). SQGEF integrates knowledge from diverse datasets that vary in granularity, time span, and observation variables. It learns unified representations for multi-granularity entities, including those not present during training. The key contributions of this work are threefold: (1) a novel method for integrating heterogeneous datasets; (2) a unique embedding approach for entities of different granularities; and (3) comprehensive experiments that demonstrate the method's effectiveness and robustness across various regions and granularities.

## update after rebuttal

The author's detailed response has largely addressed my initial concerns, so I have updated my assessment to positive.

**Claims And Evidence:**

The claims are partially supported. How combining multi-granularity datasets benefits forecasting performance seems to be missed.

**Essential References Not Discussed:**

The coverage of references is comprehensive.

**Experimental Designs Or Analyses:**

The experimental design is reasonable but could be improved.

**Methods And Evaluation Criteria:**

The method is theoretically effective and feasible.

**Other Comments Or Suggestions:**

See above.

**Other Strengths And Weaknesses:**

**Strengths:**

- The idea of segment quadtree is novel. And the whole framework is technically sound.
- The experiment covers different regions, data scarcity, and backbones.

**Weaknesses:**

- Additional time consumption of applying the proposed framework should be provided.
- The theoretical analysis is sketchy. More details should be provided, e.g., the calculation of bounds.
- The scarcity of different datasets should be given.

**Questions For Authors:**

See above.

**Relation To Broader Scientific Literature:**

The idea of segment quadtree is novel to me. I haven't seen similar technology in existing literature.

**Theoretical Claims:**

A rough theoretical analysis is given, while missing some details about calculation procedure.

---

> ### Author Rebuttal · Authors · 2025-04-01
>
> Thank you for your valuable comments on our work. We greatly appreciate your feedback, including the detailed suggestions for analysis and clarification. However, due to space constraints, we cannot include all the content in the main text. Since URLs are not permitted except for figures, we have included the updated PDF with detailed proofs in the supplementary materials (uploaded to an anonymous repository). All referenced appendices are provided. We sincerely hope you will review these materials and reconsider our score.
>
> # Benefits of Multi-granularity
> The ablation study w/o SE empirically shows that combining multi-granularity datasets is effective, as performance drops noticeably when this component is removed. To shed more light, we explain below the intuition behind why this approach boosts forecasting performance.
>
> In SQGEF, combining multi-granularity datasets enhances forecasting by allowing each Quadtree node to capture info from its own and higher granularity levels. This hierarchical integration occurs during training, where a node’s queried embedding is computed by **fusing its own embedding with those of all child nodes** stored in the tree, as shown in Equation 1. When a high-level node gets supervision from a coarse-granularity dataset, its child nodes are updated simultaneously. This top-down approach ensures lower-level nodes benefit from the richer, global info in higher-level datasets.
>
> This mechanism provides key benefits. First, it lets lower-level nodes gain global interactions beyond their limited local data. Second, it supports cross-scale training: lower-level nodes are improved using higher-level dataset supervision, allowing the model to capture nested relationships across granularities. Thus, SQGEF builds a richer representation of geographical entities, enhancing generalization to unseen entities or data-scarce regions.
>
> # W1
> SQGEF introduces only constant additional computational complexity. During training, it maintains the same space and computational complexity as the naive method while capturing multi-granularity relationships for significant performance gains. During inference, SQGEF achieves lower complexity than the naive approach, ensuring superior scalability and efficiency.
>
> Due to space constraints, the detailed analysis is available in the complexity analysis (Rebuttal, Reviewer cJrD) or Appendix A.8.
>
> # W2
> We have added a comprehensive proof in Appendix A.5. Due to space constraints, we only summarize the key conclusions here.
> From the Rademacher Complexity Theorem, for any $\mathcal{H}$, the generalization error bound is $L(h) \leq L^S(h) + 2\hat{\mathcal{R}}_S(\mathcal{H}) + 3\sqrt{\frac{2\ln(2/\delta)}{n}}$, where $\hat{\mathcal{R}}_S(\mathcal{H})$ is the empirical Rademacher complexity. For Segment Quadtree Embedding, $\hat{\mathcal{R}}S(\mathcal{H}{\text{seg}}) \leq C_2 \cdot \frac{|W|^{L+1} \sqrt{p'} \sqrt{m d}}{\sqrt{n}}$, with $m = O(\log k)$ and $p' \propto m d$, giving a tighter bound for Segment Quadtree Embedding.
>
> # W3
> We first clarify what data scarcity means in our work and then provide statistical evidence to support it. Due to space constraints, we have added detailed description in Appendix A.6.
>
> In our paper, data scarcity refers to forecasting targets in the test set being unseen during pretraining, with no historical data available at that stage. This severely limits direct modeling of their relationships. To tackle this, SQGEF uses a large volume of heterogeneous datasets from related regions or entities, despite lacking test target historical records.To quantify this scarcity, we compare the data volume between the pretraining and test stages across three metrics: total data points, number of temporal points, and temporal duration.
>
>
> |Experiment|Pretrain Set Data Points|Test Set Data Points|Pretrain Set Time Points (Span)|Test Set Time Points (Span)|
> |-|-|-|-|-|
> |China Province|8,651,704|37,062|581 (22 years)|1,278 (2 years)|
> |Europe Country|8,650,752|39,618|528 (22 years)|1,278 (2 years)|
> |China City|8,687,814|962|1,806 (22 years)|37 (37 years)|
>
> From this table, it is evident that the test set contains significantly fewer data points than the pretraining set across all experiments. For China Province and Europe Country, the test set’s temporal duration is notably short(2 years). In the China City, test set has an extremely limited number of temporal records(37 points), indicating sparse sampling. Collectively, these statistics highlight the severe data scarcity in the test set, both in terms of quantity and coverage.
>
> Moreover, Table 2 in our paper shows that this scarcity goes beyond data volume to informational richness. Models using only the scarce historical data of forecasting targets consistently lag behind SQGEF, which combines this limited target data with richer, more plentiful data from related grids and entities during pretraining.

---

### Official Review · Reviewer_cJrD · 2025-03-17

**Overall Recommendation:** 4

**Summary:**

This paper aims to integrate diverse heterogeneous datasets to address the data scarcity problem in scientific research. The authors propose the Segment Quadtree Geographical Embedding Framework with a novel data structure called the Segment Quadtree. The framework employs two learning strategies: capture interactions at multiple levels from grid datasets, and extract nested relationships and human-defined boundary information from entity datasets. As a result, SQGEF enhances spatio-temporal forecasting for geographical entities across various regions and granularities, even for entities unseen in the training set.

## Rebuttal summary

Data scarcity has been a fundamental issue in scientific research. To resolve this, this paper proposes integrating diverse heterogeneous datasets with a novel Segment Quadtree Geographical Embedding Framework. This framework is verified with comprehensive experiments. My initial concerns about experimental designs have been reasonably addressed after the rebuttal, so I have raised my score from 3 to 4.

**Claims And Evidence:**

The paper makes two main claims: 1. Heterogeneous datasets from different studies can provide complementary insights into the same underlying system. 2. SQGEF is designed to learn unified representations for multi-granularity entities, including those absent during training.

The authors support these claims by using datasets from different sources within the same region. The improved performance on forecasting targets that were absent in the training data validates these claims. Overall, the claims are well-supported and convincing.

**Essential References Not Discussed:**

No

**Experimental Designs Or Analyses:**

The overall setting is comprehensive. The authors mainly support the claims from different perspectives. However, the baseline is outdated. The paper should include some more advanced baselines.

**Methods And Evaluation Criteria:**

Yes, they make sense.

**Other Comments Or Suggestions:**

No.

**Other Strengths And Weaknesses:**

Strengths:
1. The paper addresses a meaningful and challenging problem: integrating heterogeneous datasets to tackle data scarcity in scientific research.
2. The methodology is well-motivated, and the Segment Quadtree data structure is innovative and interesting.
3. The paper provides a comprehensive set of experiments, addressing multiple research questions and demonstrating the framework's effectiveness across various scenarios.

Weakness:
1. The baselines used in the experiments are really outdated. There are many more advanced baselines available.

2. The paper lacks a thorough analysis of the framework's efficiency and scalability. For example, there is no discussion of computational complexity, training time, or memory usage, which are critical for real-world applications.

**Questions For Authors:**

1. What is the performance of the framework on more advanced baselines, both for time series models and spatio-temporal models?

2. What is the computational complexity of the model? Can it generalize to larger datasets or higher-resolution grids?

**Relation To Broader Scientific Literature:**

This paper addresses the data scarcity problem in the scientific data domain, which has broad applications in fields such as satellite data analysis, and remote sensing.

**Theoretical Claims:**

I reviewed the proof in Section 4, and it is correct.

---

> ### Author Rebuttal · Authors · 2025-04-01
>
> Thank you for your thorough review! Your suggestion to include additional experiments and analysis will further strengthen our paper.
>
> # outdated baselines
>
> Thank you for your suggestion! We have updated our experiments to include two state-of-the-art baselines: TimeXer (NeurIPS 2024) and iTransformer (ICLR 2024). The results demonstrate that our framework significantly enhances the performance of these cutting-edge baselines, underscoring its robustness and adaptability.
>
> |China Province||||Europe Country||||China City|||
> |---|---|---|---|---|---|---|---|---|---|---|
> |Method|MSE|MAE|Method|MSE|MAE|Method|MSE|MAE|
> |TimeXer|0.4027±0.0000|0.4894±0.0000|TimeXer|2.3852±0.0066|1.1684±0.0005|TimeXer|8.1542±0.7333|1.3078±0.0022|
> |w/Seg|0.3569±0.0002|0.4596±0.0001|w/Seg|2.2058±0.0022|1.1266±0.0002|w/Seg|7.3635±0.3326|1.2838±0.0026|
> |iTransformer|0.4772±0.0064|0.5004±0.0043|iTransformer|2.5433±0.0000|1.1808±0.0000|iTransformer|8.5086±0.4185|1.3474±0.0018|
> |w/Seg|0.4319±0.0001|0.4790±0.0001|w/Seg|2.2671±0.0001|1.1454±0.0000|w/Seg|7.8494±0.0765|1.3240±0.0015|
>
> # complexity analysis
>
> Thank you for your insightful review! We agree that including an efficiency analysis strengthens our paper, and hightlights the effectiveness of our method that we only introduces a constant-level additional complexity to achieve significant performance increase.
>
> **Space Complexity:** Both SQGEF and naive grid embedding method have a space complexity of O(N) (where N is the number of grid cells), SQGEF introduces only a constant-level overhead to store multi-level interaction information. In our Segment Quadtree, the total number of nodes is N (root level) + (1/4)N (next level) + ... + (1/4)^h N, where h = ⌈log₄N⌉ is the tree height. This geometric series sums to O((4/3)N), which remains linear in N. Thus, our approach scales efficiently even with higher-resolution grids, incurring minimal additional storage costs.
>
> **Training Stage Time Complexity:** While both SQGEF and the naive grid embedding method have a computational complexity of *O(N²)* (where *N* is the number of grid cells), SQGEF adds only a constant overhead to process multi-granularity data. While SQGEF models interactions across all granularity levels, the additional computational cost is minor relative to the substantial performance gains it delivers. The complexity comprises two parts: (1) constructing a graph to capture spatial relations between grids, and (2) computing temporal relations across timestamps. For the graph, the naive method’s complexity is O(N²). In SQGEF, this becomes O(N² + (1/16)N² + ... + (1/16)^h N²) across tree levels, summing to O((16/15)N²), a slight increase over O(N²). Temporal relations are treated as a constant T, as no additional temporal computation is required beyond the input sequence. This modest overhead enables rich multi-level interactions, significantly enhancing forecasting accuracy, as evidenced by our experimental results.
>
> **Inference Stage Time Complexity**: During inference, SQGEF outperforms the naive grid embedding method in efficiency. Using the segment tree query method [1]for a region [qX₁, qX₂] × [qY₁, qY₂], we recursively check overlaps with the current node’s region [X₁, X₂] × [Y₁, Y₂]. With a tree height of O(log₄N), and up to 4 nodes visited per level (if the query spans quadrants), the total nodes visited is bounded by 4·O(log₄N) = O(log N) (since log₄N = (1/2)log₂N). This contrasts with the naive grid method’s O(N) complexity, making our approach far more efficient.
>
> **Summary:** During training, SQGEF maintains *O(N)* space complexity and *O(N²)* computational complexity, identical to the naive method, enabling the capture of multi-granularity relationships that yield significant performance improvements. In inference, our method achieves O(log N ) complexity, versus O(N) for the naive approach, ensuring superior scalability and efficiency. These trade-offs make SQGEF highly practical and advantageous, especially for numerous downstream spatio-temporal forecasting tasks, where its fast inference speed delivers substantial benefits across a wide range of applications.
>
> [1] Lee, D. T., & Preparata, F. P. (1984). *Computational Geometry: A Survey*. IEEE Transactions on Computers.

---

### Decision · Program_Chairs · 2025-05-01

**Decision:**

Accept (poster)

**Comment:**

The paper proposed a novel framework, the Segment Quadtree Geographical Embedding Framework (SQGEF), to address the challenge of data scarcity in geographical scientific datasets. SQGEF integrates knowledge from diverse datasets that vary in granularity, time span, and observation variables. The reviewers agree that the proposed ideas are interesting and useful. Some concerns related to experimental evaluations and paper presentations are addressed during rebuttal. The authors are suggested to incorporate these revisions in the manuscript.